# Age-Related Alterations in Immune Contexture Are Associated with Aggressiveness in Rhabdomyosarcoma

**DOI:** 10.3390/cancers11091380

**Published:** 2019-09-17

**Authors:** Patrizia Gasparini, Orazio Fortunato, Loris De Cecco, Michela Casanova, Maria Federica Iannó, Andrea Carenzo, Giovanni Centonze, Massimo Milione, Paola Collini, Mattia Boeri, Matteo Dugo, Chiara Gargiuli, Mavis Mensah, Miriam Segale, Luca Bergamaschi, Stefano Chiaravalli, Maria Luisa Sensi, Maura Massimino, Gabriella Sozzi, Andrea Ferrari

**Affiliations:** 1Tumor Genomics Unit, Department of Research, Fondazione IRCCS Istituto Nazionale dei Tumori, Via Venezian 1, 20133 Milan, Italy; orazio.fortunato@istitutotumori.mi.it (O.F.); mattia.boeri@istitutotumori.mi.it (M.B.); mensah.mavis@istitutotumori.mi.it (M.M.); miriam.segale@istitutotumori.mi.it (M.S.); gabriella.sozzi@istitutotumori.mi.it (G.S.); 2Integrated Biology Platform, Department of Applied Research and Technology Development, Fondazione IRCCS Istituto Nazionale dei Tumori, Via Venezian 1, 20133 Milan, Italy; maria.ianno@istitutotumori.mi.it (M.F.I.); andrea.carenzo@istitutotumori.mi.it (A.C.); matteo.dugo@istitutotumori.mi.it (M.D.); chiara.gargiuli@istitutotumori.mi.it (C.G.); marialuisa.sensi@istitutotumori.mi.it (M.L.S.); 3Pediatric Oncology Unit, Fondazione IRCCS Istituto Nazionale dei Tumori, Via Venezian 1, 20133 Milan, Italy; michela.casanova@istitutotumori.mi.it (M.C.); luca.bergamaschi@istitutotumori.mi.it (L.B.); stefano.chiaravalli@istitutotumori.mi.it (S.C.); maura.massimino@istitutotumori.mi.it (M.M.); andrea.ferrari@istitutotumori.mi.it (A.F.); 4Department of Diagnostic Pathology and from the Laboratory of Medicine of our Institute, Fondazione IRCCS Istituto Nazionale dei Tumori, Via Venezian 1, 20133 Milan, Italy; giovanni.centonze@istitutotumori.mi.it (G.C.); massimo.milione@istitutotumori.mi.it (M.M.); paola.collini@istitutotumori.mi.it (P.C.)

**Keywords:** rhabdomyosarcoma, pediatric tumors, adolescents, miRNA, gene expression, immune contexture

## Abstract

Adolescents and young adults (AYA) with rhabdomyosarcoma (RMS) form a subgroup of patients whose optimal clinical management and access to care remain a challenge and whose survival lacks behind that of children diagnosed with histologically similar tumors. Understanding the tumor biology that differentiates children from AYA-RMS could provide critical information and drive new initiatives to improve the final outcome. MicroRNA (miRNA) and gene expression profiling (GEP) was evaluated in a RMS cohort of 49 tumor and 15 non-neoplastic tissues. miRNAs analysis identified miR-223 over-expression and miR-431 down-regulation in AYA, validated by Real-Time PCR and miRNA in situ hybridization (ISH). GEP analysis detected 793 age-correlated genes in tumors, of which 194 were anti-correlated. *NOTCH2*, *FGFR1/2* were significantly down-modulated in AYA-RMS. miR-223 was associated with up-regulation of epithelial mesenchymal translation (EMT) and inflammatory pathways, whereas miR-431 was correlated to myogenic differentiation and muscle metabolism. GEP showed an increase in genes associated with CD4 memory resting cells and a decrease in genes associated with γδ T-cells in AYA-RMS. Immunohistochemistry (IHC) analysis demonstrated an increase of infiltrated CD4, CD8, and neutrophils in AYA-RMS tumors. Our results show that aggressiveness of AYA-RMS could be explained by differences in microenvironmental signal modulation mediated by tumor cells, suggesting a fundamental role of immune contexture in AYA-RMS development.

## 1. Introduction

Rhabdomyosarcoma (RMS) is a pediatric malignancy which originates from mesenchymal progenitor cells located in muscle tissue, even if its cell of origin is still debatable. Histologically, RMS is characterized by two main subtypes with distinct molecular profiles: embryonal (ERMS) and alveolar (ARMS). RMS is the most frequent soft tissue sarcoma (STS) in children and adolescents, but it can also occur in adults. Patients’ outcome is influenced by certain prognostic variables (i.e., histology, local and distant invasiveness, tumor site and size) that are currently considered for stratifying patients and guiding treatment strategies and intensity [1]. Among other prognostic features, patient’s age has emerged to be a factor that significantly impacts survival in RMS patients. The epidemiological EUROCARE-5 working group (study period: 2000–2007) reported a 66.6% 5-year overall survival among 0 to 14 years old patients (PEDS-RMS), as opposed to 38% for patients 15 to 39 years of age (AYA-RMS) [2].

The rationale for the adolescents’ survival gap is likely multi-factorial. It has been suggested that variables related to clinical management and organization of care do play an essential role. Indeed, it has been shown that adolescents and young adults (hereafter, AYA) subgroup had a significantly higher prevalence of unfavorable features (e.g., alveolar subtypes, unfavorable sites, advanced disease at onset), a considerable delay in diagnosis [3], and a reduced likelihood to be enrolled in the national cooperative pediatric protocol [4]. Moreover, some studies proposed that the quality of treatment administered, in terms of its adherence to the principles adopted in pediatric protocols, may influence patient outcomes [5,6]. Treating adults with a pediatric-type strategy is not sufficient to achieve similar results obtained in children. Inadequate treatment is certainly an issue, but a more aggressive biology of adult RMS might play a pivotal role in the different outcome according to age [6]. Nonetheless, tumor biology related factors could also be responsible for the prognostic gap between children and AYA. Despite growing knowledge on the mutational landscape on RMS [7,8], evidence and observations do not address age-related differences and aggressiveness of the disease. Accordingly, an enhanced integrated, comprehensive approach of the genomic aspects contributing to tumorigenesis in pediatric (PEDS) and AYA-RMS may identify different targets and/or pathways that discriminate the two age groups, thus improving therapeutic intervention and regimen, and ultimately guide clinical management.

MicroRNAs (miRNAs) are small non-coding RNAs that regulate gene expression through translational repression or degradation of their targets [9]. miRNAs are deregulated in different diseases, such as cancer, hepatitis, and metabolic disease [10]. Based on their function, miRNAs may be considered as “oncomir” when their over-expression in tumor inhibits a tumor suppressor gene, for instance, a mir-17-92 cluster in lung cancer [11]. On the other hand, a “tumor suppressor miRNA”, such as let-7a, is down-regulated in cancer cells, and its loss causes aberrant expression of an oncogene [12]. Evidence reports that both myomiRs, which are muscle enriched miRNAs, and non-myomiRs, which are expressed in muscles but also in other tissues, play a critical role in RMS development, progression, and could potentially be therapeutic targets for RMS [13,14]. In particular, miR-22, a miRNA induced in physiological normal muscle differentiation, is down-regulated in RMS, and its replacement blocked tumor growth and dissemination in pre-clinical in vivo models [14]. Moreover, as myomiRs are essential for skeletal muscle differentiation, their down-regulation may be one of the factors responsible for their dedifferentiated phenotype in RMS [13]. Furthermore, miR-206 was described as a skeletal muscle miRNA that controlled the proliferation of muscle cells through Pax7 down-regulation [15]. Regarding non-myomiRs, the miR-17-92 cluster is over-expressed in RMS and could be responsible for RMS aggressiveness [16]. Another miR linked to metastatic dissemination of RMS is miR-185 that modulates Six1 [17]. While gene expression profiling (GEP) in RMS provides survival prediction signatures and define molecular classes [18,19], no studies focused on the description of age-related biological targets and pathways have been so far performed.

The transition from childhood to adult state is associated with physical and emotional turmoil, all factors that threaten the ability of adolescents to become healthy and productive adults. From a biological standpoint, substantial differences, such as hormonal imbalances and muscle development, characterize adolescents compared to pediatric patients [20,21,22]. Interestingly, epidemiological studies have shown that inflammation predisposes individuals to various types of pediatric cancer. It is estimated that underlying infections and inflammatory responses are linked to 15% to 20% of all deaths from cancer worldwide [23].

Here, we propose a broad exploration of pediatric and AYA-RMS by integrating miRNAs and gene expression assessment to identify putative miRNAs expression, target genes modulation, pathways, and biological mechanisms that could discriminate children from AYA-RMS pathogenesis. We hypothesize that these analyses could unveil promising biomarkers or novel therapeutic agents with clinical utility for AYA-RMS, to guide clinical management and ultimately, patients’ survival.

## 2. Results

### 2.1. Clinico-Pathological Features of the Studied RMS Series

The retrospective clinical series analyzed consisted of 49 RMS (28 pediatric and 21 AYA) samples, whose clinical-pathological description is summarized in Table 1.

Overall, the clinical cohort used in this study realistically represented RMS malignancy considering all different clinico-pathological characteristics. Patient’s survival was evaluated according to age, Intergroup Rhabdomyosarcoma Staging (IRS) staging, and histology. A non-significant separation of Kaplan-Meier survival curves was observed between PEDS-RMS and AYA-RMS patients (*p* = 0.1458, Figure 1A and Appendix A). IRS IV grade reached a hazard ratio (HR) of 3.5 (*p* < 0.0001, Figure 1A and Appendix A). As already described, a worst prognosis was observed in ARMS subtypes compared to ERMS (HR = 2.8, *p* = 0.0187 Figure 1A). Histologically, of 49 RMS, 29 were diagnosed as embryonal while 20 alveolar, of which 10 resulted positive for the *PAX* fusion transcript, five negative and five were classified as not performed (insufficient material to investigate their *PAX* fusion status). As expected fusion-positive patients have a worst survival (*p* = 0.1089 Figure 1A). The Kaplan-Meier’s curve (Figure 1A) representing our cohort, showed a worse outcome of AYA-RMS compared to PEDS-RMS, although AYAs’ outcome is influenced by the careful clinical management of AYA patients in our Institution. Finally, morphological and histological features characterizing ERMS and ARMS were similar in all RMS tissue samples regardless of the age of patients (Figure 1B).

### 2.2. Age-Related miRNA Expression in RMS

According to Spearman’s rank correlation tests and by imposing a significance level α < 0.05, we detected in tumor samples 39 miRNAs out of 1710 miRNAs to be positively correlated (correlation coefficient ρ > 0.3) with age in tumor specimens, whereas 20 miRNAs were anti-correlated (ρ < −0.3), (Figure 2A and Appendix A). Considering non-neoplastic tissue samples imposing α < 0.05, we found 22 positive correlated (ρ > 0.3) miRNAs, nine of which were anti-correlated with age (ρ < −0.3) (Appendix A and Appendix A). To identify possible candidate miRNAs deregulated in the tumor but not in non-neoplastic tissue, 10 miRNAs (Appendix A) were selected considering the nominal *p*-value < 0.05 in the tumor and a *p*-value ≥ 0.4 in normal tissue, meaning those statistically significant only in tumors. Interestingly, nine out of 10 of these miRNAs (Appendix A) were described to be involved in myogenesis, muscle differentiation, age-related issues, stem cells, metastasis, and hormonal changes [11]. Results obtained from the microarray analysis were also validated by Real-Time PCR. This analysis was performed in only 53 samples with sufficient RNA material (48 from tumor, 5 from non-neoplastic tissues) using custom-made microfluidic cards. Interestingly, miR-223 was up-regulated in AYA-RMS compared to PEDS-RMS (Figure 2B, *p* = 0.031) and also considering age as a continuum (Figure 2B, Pearson r = 0.47, *p* = 0.033), whereas miR-431 expression was up-regulated in pediatrics samples (Figure 2C, Pearson r = −0.47, *p* = 0.032).

miR-223 was detected to be deregulated in cancer-related pathways, such as FOXO signaling, and in signaling pathways involved in microenvironmental mechanisms, such as hypoxia and AMPK/mTOR (Figure 2B). Kyoto Encyclopedia of Genes and Genome (KEGG) pathways analysis revealed that miR-431 regulates cell cycle genes, such as *MCM6, CDC45* and *ORC5*, glycosaminoglycans degradation (*GUSB*), and drug metabolism (*GUSB* and *NAT1*) (Figure 2C). To validate the expression of miRNAs deregulated in RMS tissues, miRNA in situ hybridization (ISH) on a subgroup of RMS slides was performed. We confirmed that miR-223 expression is higher in AYA-RMS tumor tissues, both in tumor and stromal cells, compared to PEDS-RMS (Figure 2B; n = 3). As for miR-431, we confirmed a decreased expression in tumor cells of AYA-RMS as opposed to PEDS-RMS (Figure 2C; n = 3).

### 2.3. Age-Related Gene Expression Assessment

By imposing the same thresholds utilized for miRNA analysis, we detected 793 age-correlated genes in tumor samples, of which 194 were anti-correlated (ρ < −0.3 and *p*-value < 0.05) and 656 age-correlated genes in non-neoplastic samples, of which 287 were anti-correlated (ρ < −0.3 and *p*-value < 0.05) (Figure 3A and Appendix A). *NOTCH2* was identified as one of the most significant deregulated gene, specifically down-modulated in AYA-RMS tumors with an important role in RMS metastatic dissemination (Appendix A). Other genes identified as significantly modulated in AYA-RMS were *Fibroblast Growth Factor Receptor 1 and 2 (FGFR1* and *FGFR2*) described as critical mediators of RMS oncogenesis, and MAPK proteins crucially important in RMS. Considering age, several genes responsible for inflammatory protein secretion/activation (*KLK3, EDN3*), cell-adhesion (*PI4KA, AHSAI1*) and cancer-related proteins (*KLK3, FGFR1/2, PRKACA,* and *RAC2*) were also modulated in AYA-RMS (Appendix A).

Gene Set Enrichment Analysis (GSEA) pathway analysis of RMS tumor samples revealed an up-regulation of cancer-related pathways, such as epithelial mesenchymal translation (EMT) and KRAS signaling (Normalized Enrichment Score (NES) intensity: 1.51 and 1.73, respectively) reflecting the aggressiveness of AYA-RMS. Interestingly, pathways linked to inflammation, such as TNF-α, IL-2, TGF-β (NES intensity: 2.04, 1.70 and 1.65, respectively), were up-regulated with age, suggesting that microenvironment plays a crucial role in RMS tumorigenesis. Altogether, our gene expression data revealed a potential interplay between hypoxia, reactive oxygen species (ROS), and inflammation in RMS development. The only pathway that resulted in down-regulation was myogenic differentiation (NES intensity: −1.41), suggesting that AYA-RMS could be less differentiated and more aggressive compared to PEDS-RMS (Figure 3B). In non-neoplastic tissues, we observed up-regulation of the same pathways involving inflammatory proteins and signals confirming the importance of an inflamed microenvironment in early phases of RMS carcinogenesis (Figure 3B). Notably, gene expression data in non-neoplastic tissues sustained the idea that the block of myogenic differentiation could be a relevant finding for RMS development.

To confirm the results obtained with GSEA analysis in tumor samples, we searched for the most relevant genes involved in the deregulated pathways described above, discovering that *SERPINE, TNFAIP3, IRF1, PLAUR, IL6, ID2,* and *BMP2* were among the most recurrent genes. Using Real-Time PCR, in a subgroup of AYA- and PEDS-RMS (n = 10 for each group) we demonstrated that *SERPINE, TNFAIP3*, and *IRF1* levels were higher in AYA-RMS samples compared to PEDS-RMS (AYA-RMS fold increase vs. PEDS-RMS: *SERPINE* 4.0; *TNFAIP3* 4.5; *IRF* 1.9) (Figure 3C and Appendix A).

### 2.4. Integration of miRNA Expression and GEP Data

miRNA and mRNA expression data were combined together, as described in Material and Methods. Thirty-nine target genes were identified as down-modulated by miR-223 in RMS, among which were *FOXO1, FOXO3, PAX3*, and *PAX4*. Three genes resulted in the most anti-correlated: *CDS1, SP3,* and *ARTN*. In RMS tumor tissues, integration analysis showed an up-regulation of EMT pathways correlated with miR-223 levels speculating that regulation of tumor suppressor Fbxw7 could be responsible for this pro-tumorigenic effect (Figure 4A and Appendix A). Our data demonstrated that miR-223 over-expression with age correlates with the enhanced inflammatory response through down-modulation of *FOXO3* and concomitant *NFKB* modulation. miR-223 regulates several targets, such as NLRP3, NFIA, and PARP1, that are involved in the induction of apoptotic mechanisms observed in AYA-RMS tumor tissues. As expected, miR-223 was able to down-modulate *PAX4* and *PAX6* expression, both fundamental genes for skeletal muscle system development and differentiation.

The same integrated analysis was performed considering miR-431 expression, revealing 49 target genes. Among these, *GLIS3* and *ZNF245* are the most anti-regulated ones (Figure 4B and Appendix A). miR-431 appears involved in cell cycle control, through regulation of cyclin-dependent kinases, such as *CDKL2* and *CKAP2L*. Interestingly, miR-431 expression has a crucial role in myogenesis and E2F targets, as shown in Figure 4B, through direct regulation of *FOXP1, IKBIP*, and a bone-derived signaling molecule Indian Hedgehog (IHH). A mitochondrial protein, *FEM1A*, is among the genes deregulated by miR-431. This miRNA is also crucial for the metabolism’s mechanisms by inhibiting *DSEL* expression.

Overall, our integrated analysis revealed that both miR-223 and miR-431 are implicated in immune-system regulation through several crucial mechanisms, such as recruitment of immune cells, with down-modulation of *SIGLEC8* and *IL-7* by miR-431 or *Leukemia inhibitory factor (LIF*) by miR-223.

### 2.5. Assessment of Specific Immune Cell Tumor Infiltration

Since miRNAs and GEP investigations highlighted the relevance of inflammation and immune contexture in AYA- and PEDS-RMS, we utilized CIBERSORT, an analytical tool developed by Newman et al., to provide an estimation of the abundances of the immune composition of RMS tumor sample using gene expression data. Notably, this approach revealed a 7.5-fold increase of CD4 resting memory in AYA-compared to PEDS-RMS without any modulation in the non-neoplastic tissues (Figure 5A). γδ T-cells were the only immune population down-modulated in tumor considering age (60% of reduction in AYA-RMS). Moreover, an increase of neutrophils infiltrating both in tumor and non-neoplastic tissues was observed in AYA-RMS (Figure 5A).

An increased expression in AYA-RMS compared to PEDS-RMS of immune cells markers, such as CD8, CD4 for lymphocytes, CD14 for monocytes and CD68 for M1 macrophages, was confirmed through Real-Time PCR (Figure 5B), suggesting lymphocytic and macrophagic infiltration in RMS tissues. To assess the involvement of immune infiltrated cells in RMS development, we performed immunohistochemistry (IHC) analysis in a restricted group of RMS tissues (10 PEDS-RMS and 11 AYA-RMS). Interestingly, AYA-RMS immune cells infiltrated tumor burden and replaced tumor cells, suggesting an important role of these cells in sustaining tumor growth. Of note, lymphoid cells distribution appears as marker-driven: CD3, CD8, HLA-1, and HLA-DR are distributed (as indicated by arrows Figure 5C) in such a way that they push and press the surrounding tissue otherwise CD4, CD68, and myeloperoxidase (encircled in Figure 5C) are distributed among the negative cells giving an intermingled pattern of staining. Conversely, PD-L1 expression was absent in tumor cells, whereas immune infiltrating cells were positive in both PEDS-RMS and AYA-RMS samples (Figure 5C).

In all AYA-RMS samples, an increase of immune cells outside the tumor area and infiltrating the tumor burden was observed and quantified (Figure 5C and Appendix A), suggesting a potential role of immune cells in the modulation of the aggressiveness of this specific age group of RMS.

## 3. Discussion

AYA with cancer form a subgroup of patients whose optimal clinical management and best possible access to care remain a challenge. In particular, AYA-RMS is reduced to histologically similar pediatric tumors [24,25]. Understanding the age-related tumor biology factors will provide critical information and guide new initiatives to improve their final outcome. Our miRNA study demonstrated that two miRNAs are differentially expressed in RMS tumor tissues according to age, but not in normal samples. In particular, we found that miR-223 is over-expressed in AYA-RMS, whereas miR-431 is over-expressed in PEDS- ones.

Numerous studies have reported miR-223 involvement in several malignancies as either a tumor suppressor or an oncogenic regulator [26,27,28]. Of particular interest is a work by Wu et al. demonstrating involvement of miR-223 in regulating *FOXO1* expression and cell proliferation. The FOXO pathway is critically implicated in RMS pathogenesis as the alveolar histology is characterized by recurrent chromosomal rearrangements involving FOXO1: t(2;13)(q35;q14), reciprocal balanced translocations *PAX3-FOXO1* and t(1;13)(p36;q14) *PAX7-FOXO1*, and other rare variants always involving FOXO1 [26]. Several works showed that miR-223 is involved in hematopoietic lineage differentiation and function with a critical role in the regulation of inflammation and immune cell recruitment [29]. These data highlight the critical importance of unveiling miR-223 its biological role in age-related RMS.

Extended literature on miR-431 reported its crucial role in myogenic differentiation, suggesting that age-associated miR-431 expression is required for the maintenance of the myogenic capability in myoblasts. Lee et al. described an improved myogenic differentiation when miRNA-431 is overexpressed in old myoblasts, but suppressed their myogenic capability when knocked-down in young myoblasts. miR-431 directly binds to 3’untranslated region (UTR) of *Smad4* mRNA and decreases its expression resulting in defective myogenesis [30]. Similarly, miR-431 is reported to regulate the myogenic capacity of human skeletal myoblasts and skeletal muscle stem cells by fine-tuning PAX7 levels [31].

Gene expression analysis of tumor samples revealed 793 age-correlated genes in RMS, of which 194 were anti-correlated. Interestingly, the most down-modulated gene is *NOTCH2.* The *NOTCH* family is described as crucial for the maintenance of quiescence of muscle stem cells in adults, suggesting a possible role of stem cells in the aggressiveness of AYA-RMS [32]. One of the up-regulated genes modulated in AYA-RMS is *GREM1*, that is known as a biomarker for cancer-associated myofibroblasts, with a potential role of micro-environmental modulation in AYA-RMS [33]. Other deregulated genes, such as *CLUH*, were described with an active role in mitochondrial metabolism [34] as *DUSP5* in the modulation of MAPK/ERK pathways in the muscle during acute exercise [35]. Notably, we observed down-modulation of *FGFR1* that is associated with skeletal muscle loss and disruption of myofiber organization [36].

miRNA and gene expression data were integrated to identify target genes for a single miRNA. We focused on integrating networks for our validated miR-223 and miR-431. For miR-223, we identified *SP3, CDS1*, and *ARTN* as the most anti-correlated genes, all involved in the development of several types of tumors. Notably, Sp3 has been involved in pancreatic cancer cells by its ability to suppress p27 expression through interaction with GC-rich promoter elements [37] and in breast cancer by its role in accelerating tumor cell growth acting as a repressor of *TGFB* [38]. Of note, upregulation of Sp3 expression in soft tissue sarcoma (STS) compared to normal connective tissue has been confirmed in the web database ONCOMINE. Considering that Sp3 is expressed at higher levels in STS and transactivates the *AFAP1L1* gene, targeting *Sp3* could be a powerful approach to treating advanced STS [39]. For miR-431, 49 target genes were considered. Among these, *GLIS3* and *ZNF245* are the most anti-regulated ones. Our integrated data also show an association of miR-431 with myogenesis mechanisms, supporting literature that describes the modulation of myostatin, muscle growth, and regeneration by miR-431 [30].

Genomic features of tumors and their microenvironments represent a source of promising candidates for predictive and prognostic biomarkers. The sarcoma immune microenvironment is characterized by a network of innate and adaptive immune cells [40]. An increase in T-cells CD4 memory resting and neutrophils infiltrating and surrounding the tumor burden was observed in our cohort of AYA-RMS compared to PEDS-RMS, suggesting AYA-RMS have a compromised/suppressed immune profile [41]. Furthermore, an increase in CD4 memory resting and neutrophils cells populations has been reported as an indication of an adverse outcome in solid tumors [41], possibly justifying the dismal outcome of AYA-RMS patients. On the other hand, PEDS-RMS presents an increase of γδ T-cells, a possible biomarker of a favorable outcome in solid tumors [41]. These γδ T-Cells displayed broad functionality by producing cytokines and chemokines and interacting with several cell types, such as epithelial cells, monocytes, neutrophils, and B cells [42]. Recruitment of these immune cell subsets in RMS microenvironment could partially explain the modulation of inflammatory signaling pathways detected by gene expression analysis [42]. These results suggest that AYA-RMS samples possess an immune-inflamed phenotype [43], supporting our hypothesis that the immune contexture in AYA-RMS is somewhat suppressed and compromised, possibly driving the age-related aggressiveness of these tumors.

Pediatric malignancies, in particular, RMS, are rare tumors, and limited cohorts and/or availability of tissue material are challenges encountered when carrying out research activities. Moreover, although clinical information was available for our cohort, the differences in overall survival were not significant between the two groups. One possible explanation could be related to the clinical management of AYA patients in our Institute, intended as different chemotherapy approaches and tumor-associated factors specific to this peculiar age group. In addition, RMS can arise nearly anywhere in the body, even in regions that normally do not have skeletal muscle. Being that miRNAs expression and modulation is tissue-specific [44], different RMS localization could challenge the identification of deregulated miRNAs.

Recently, immune-checkpoint inhibitor blockade has emerged as an effective option for the treatment of solid cancers. Differences in the immune systems of children and adults, such as patient’s Human Leukocyte Antigen (HLA) haplotype, distinctions in both immune function, and RMS biology have important practical implications for cancer immunotherapy [40]. Our data reveal a massive immune cells infiltration, in particular CD8 and CD4 cells in AYA-RMS, suggesting that age-related modifications of the immune contextures are associated with aggressiveness of RMS. Notably, PD-L1 expression in immune infiltrating cells of both PEDS and AYA-RMS suggest the possibility to target this immune checkpoint in selected patients. Further investigations are needed to better clarify the use of immune checkpoints inhibitors, combined with novel immunotherapeutic strategies to block the recruitment of immune cells for the treatment of this aggressive disease.

## 4. Material and Methods

### 4.1. Patients and Samples Selection

The study is a monocentric retrospective analysis of a cohort of consecutive cases diagnosed with RMS and treated at our Institute, recovered from our Institutional series between the years 1999 and 2016 at the Fondazione IRCCS of the Istituto Nazionale dei Tumori (INT). Seventy-six Formalin Fixed Paraffin Embedded (FFPE) tissues of primary, non-pre-treated, surgical or bioptic specimen of PEDS- and AYA-RMS were selected and retrieved from the archives of the Department of Diagnostic Pathology and from the Laboratory of Medicine of our Institute. The sample series were carefully re-evaluated by expert pathologists for abundance and adequacy of material, and 56 (30 PEDS and 26 AYA) RMS were successfully selected. In addition, pathologists identified, where possible, a non-neoplastic component (normal tissue adjacent to the tumoral area) from 15 RMS samples covering both age groups, to use as controls for bioinformatic analysis. Due to inadequacy of quality or quantity of RNA extracted from RMS samples, however, it was only possible to perform all miRNA and gene expression analyses on a total of 49 RMS (28 PEDS and 21 AYA) samples and 15 non-neoplastic tissue (9 PEDS and 6 AYA), as illustrated by Figure 6. The study was approved by the Internal Audit Committee and the Ethics Committee of our Institute (*CE N. INT 132–16*). All patients, and/or their guardians, consented to the study by signing an informed consent to research activities at the time of admission to the hospital.

### 4.2. RNA Extraction from FFPE

Total RNA was isolated from 64 (49 tumor tissues and 15 normal tissues) RMS FFPE using miRNeasy FFPE Kit (Qiagen, Valencia, CA, USA), and the procedure was automated on a QIAcube Robotic workstation. The RNA extracted was quantified on a Qubit™ dsDNA HS Assay Kit on a Qubit fluorometer (ThermoFisher, Waltham, MA, USA) while RNA integrity was assessed using Agilent RNA ScreenTape on a 4200 TapeStation, (Agilent Technologies, Santa Clara, CA, USA).

### 4.3. miRNAs Profiling

Total RNA from 64 samples was extracted, and quality controls were performed as described above. RNA was processed for miRNA profiling using miRNA Microarray System (containing ~2.000 miRNAs) with a miRNA Complete Labeling and Hyb Kit (Agilent Technology) according to the manufacturer’s instructions and as previously described [45]. miRNA expression analysis was performed using SurePrint G3 Human miRNA 8 × 60 K microarrays from Agilent Technologies. RNA was dephosphorylated, starting from 100 ng of amount, with calf intestinal alkaline phosphatase and denatured by DMSO treatment. Samples were labeled with cyanine 3-pCp using T4 RNA ligase and hybridized on miRNA array. Arrays were washed in Agilent’s Wash Buffers and scanned at a resolution of 2 mm using an Agilent DNA microarray scanner (Agilent Technologies). Primary data were collected using Agilent’s Feature Extraction software v10.7 (Agilent Technologies).

### 4.4. Real-Time PCR

A TaqMan MicroRNA Reverse Transcription Kit (ThermoFisher) was used to perform quantitative PCR Real-Time according to the standard procedure, and the relative quantification analysis was performed using the 2-ddCt method, and RNU48 as a housekeeping gene. A custom card with 55 selected miRNAs was appropriately designed and analyzed as described [46].

### 4.5. Immunohistochemistry

Expression of a panel of 8 antibodies (Appendix A) was evaluated on a restricted RMS cohort (10 PEDS-RMS and 11 AYA-RMS), selected based on the abundance of FFPE material. Briefly, sections 2.5/3 micron-thick were cut from paraffin blocks, dried, de-waxed, rehydrated, and unmasked (with Dako PT-link, EnVision™ FLEX Target Retrieval Solution, High/Low pH) (Dako, Glostrup, Denmark). All antibodies were incubated with a commercially available detection kit (EnVision™ FLEX+, Dako, Denmark) in an automated Immunostainer (Dako Autostainer System—Link 48). The status of PD-L1 protein expression was also investigated utilizing clone anti PD-L1 22C3 (Dako, Glostrup, Denmark), as previously described [47].

### 4.6. miRNAs In Situ Hybridization (ISH)

miRNAs *in situ* hybridization was performed on FFPE tissue sections, as previously described by Fortunato et al. [48]. In brief, FFPE sections (5 μm thick) were first deparaffinized in xylene, rehydrated in an alcohol descending scale and treated with Proteinase-K (Sigma-Aldrich, Saint Louis, MO, USA). Samples were hybridized with probe mixture for 2 h, in the Dako Hybridizer, at specific probe hybridization temperature (RNA Tm −30 °C) (Appendix A). For image analysis, stained sections were examined by optical microscope and scanned with Aperio Scanscope XT (Leica Biosystems, Nussloch, Germany). Slides and miRNAs signals were analyzed by expert pathologists (PC and MM).

### 4.7. Gene Expression Profiling

Total RNA from 63 RMS samples, the same case material used in miRNA profiling, was reverse-transcribed, amplified, fragmented, biotin-labeled, and then hybridized to the Affymetrix GeneChip Human Clariom S Array (ThermoFisher Scientific, Waltham, MA, USA). One RMS specimen was excluded from the analysis for an insufficient quantity of total RNA. Probe synthesis was performed using Clariom S Pico Assay; while washing and staining procedures using the GeneChip Hybridization, Wash and Stain Kit (ThermoFisher Scientific) on a GeneChipHybridization Oven 645 (ThermoFisher Scientific) according to the manufacturer’s protocol were performed. Scanning was performed on Affymetrix GeneChip Scanner 3000 TG System and primary data were collected using GeneChip™ Command Console™ (AGCC) Software (ThermoFisher Scientific). Corresponding expression array intensity (.CEL) files were processed using Affymetrix Expression Console Software (version 1.4) [49], which normalizes array signals using Signal Space Transformation (SST) and the robust multiarray averaging (RMA) algorithm. Quality controls have been implemented into the experimental workflow according to the manufacturer’s protocol to check RNA quality, probe synthesis, and hybridization performance. For gene expression analysis using Real-Time, cDNA synthesis was performed using 250 ng of total RNA. The relative quantification of the analyzed genes was performed using ready-to-use Assay-on-Demand (ThermoFisher Scientific), and human GAPDH was used as an endogenous control for normalization. The abundance of immune-components in 22 different tumors was inferred by CIBERSORT (https://cibersort.stanford.edu/) [50], a web bioinformatic application to deconvolute phenotype-specific immune cells from complex bulk gene-expression data [51].

### 4.8. Bioinformatic Analysis

The data collected were analyzed using the *R* statistical software, version 3.5.2 [52], as well as the *Bioconductor* [53,54] package *ComplexHeatmap* [55,56]. To identify genes and miRNAs whose expression levels may vary with age in RMS tumors, we used the Spearman’s rank correlation test, a versatile statistical tool not requiring either normality assumptions for data or a linear association between the two considered variables (in our case age and expression).

To compute exact *p*-values for the tests, we avoided *tiles* (i.e., when two or more equal values of a variable are assigned the same rank) in the age variable by considering the difference in days between the diagnosis and the birth date, and then converting it into years. This allowed us to pass from a discrete variable to a continuous one.
$$age\;at\;diagnosis\;\left[years\right]=\frac{\left(diagnosis\;date-birth\;date\right)\;\left[days\right]}{365\;\left[days\right]}$$


After correlation tests were conducted for every molecule (1710 miRNAs and 18,531 genes) in tumor samples choosing a significance level α = 0.05, we selected 10 miRNAs that resulted in age-correlated in tumor samples (modulus of Rho statistic at least 0.3 and *p*-value < 0.05) but not in normal ones (*p*-value ≥ 0.05), to further explore the miRNA-mRNA interactions through an integration analysis. To retrieve the genes targeted from the 10 selected miRNAs, we used *miRNET* [57], a web-based tool which collects data from 11 miRNA databases on miRNA interactions with other kinds of molecules. After filtering out target genes not present in our gene expression matrix, Spearman’s rank correlation tests at a level of significance α = 0.05 were made between the above mentioned 10 age-correlated miRNAs and their target genes obtained through *miRNET*. Finally, an interaction network consisting of negatively-correlated miRNA-gene pairs was created using the open-source software *Cytoscape (https://cytoscape.org)* [58,59].

### 4.9. miRNA-mRNA Integration Analysis

miRNA and mRNA expression data were combined in a way similar to that implemented in the MAGIA tool [60]. We obtained an integration network showing the functional relationships between the molecules through the following steps:

(1). Fifty-nine out of 1710 miRNAs were found to be age-correlated according to the Spearman’s rank correlation test and our imposed thresholds (see the above section);

(2). A subset of 10 miRNAs (deregulated in the tumor but in non-neoplastic tissue, therefore, not linked to individuals’ physiological growth) was chosen for further analyses;

(3). Target genes were obtained for these miRNAs through the web-based tool *miRNET*;

(4). Correlation tests were made between the 10 selected miRNAs and their target genes by using the expression data (i.e., among the target genes only those present in our expression matrix were considered);

(5). A miRNA-mRNA integration network consisting of the 10 miRNAs with 369 anti-correlated target genes was created using Cytoscape.

## 5. Conclusions

Functional studies of miRNAs and (target) genes of interest are necessary to fully describe biological mechanisms behind RMS pathogenesis and tumor progression. In addition to identifying several age-related miRNAs and GEP modulations, we detected changes in the modulated immune contexture in AYA-RMS, that could drive the aggressiveness of RMS in this particular age group. Further investigations will potentially unveil important information able to guide better clinical management of AYA RMS.

## Figures and Tables

**Figure 1 cancers-11-01380-f001:**
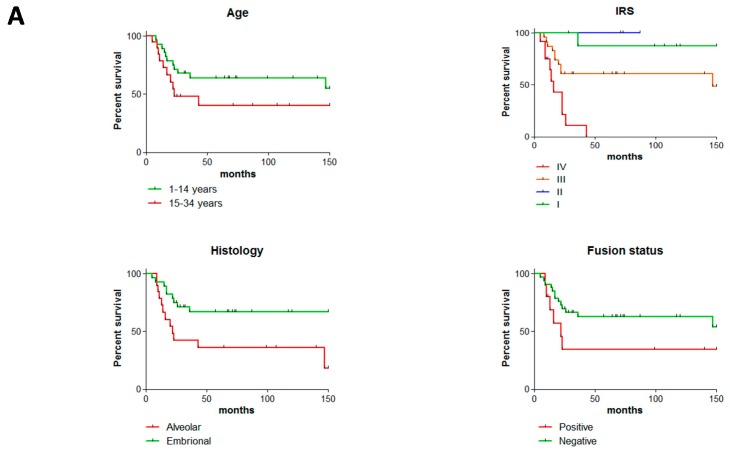
No morphology and histological differences among AYA- and PEDS-RMS. (**A**) Overall survival, IRS, Histology and Fusion status of the cohort are illustrated in Kaplan-Meier curves. (**B**) Representative images of a (**a**) Paratesticular PEDS-ERMS (13 years of age, RMS_26), (**b**) Paratesticular sample of AYA-ERMS (15 years old, adRMS_5). Similarly, images of (**c**) of PEDS-ARMS originated from extremities (8 years old, RMS_19), positive for PAX3-FOXO1 fusion transcript with identical histological features as image (**d**) AYA-ARMS originated in extremities (21 years old, adRMS_12), also positive for PAX3-FOXO1 fusion transcript.

**Figure 2 cancers-11-01380-f002:**
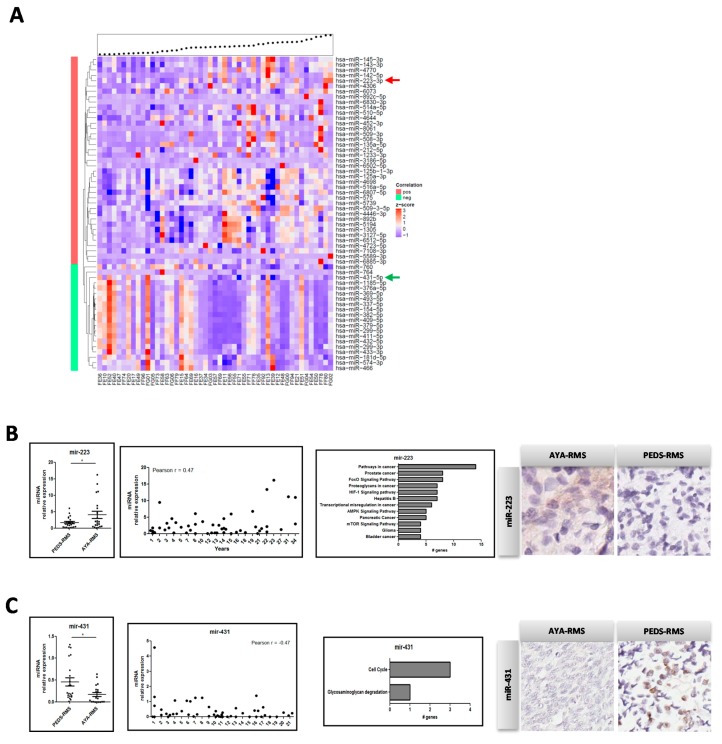
Mir-223 was upregulated and mir-431 down-regulated in AYA-RMS. **(A)** Heatmap showing the expression of the age-correlated miRNAs according to Spearman’s rank correlation test sorted by age (the black dots above the heatmap). MiRNAs corresponding to the red vertical bar are positively correlated and their expression may increase with age (n = 39), while those associated to the green bar are negatively correlated (n = 20). Red arrow indicated miR-223 whereas mir-431 with green arrow **(B)** MiR-223 expression is higher in AYA-RMS evaluated by Real Time PCR (left) and its expression is age-correlated. Graphs show KEGG pathway analysis (middle) and miRNA ISH showing tumoral expression in AYA-RMS compared to PEDS-RMS. (**C**) MiR-431 expression is up-regulated in PEDS-RMS and its expression decreases with age (left). KEGG graphs of modulated pathway by miR-431 (middle). ISH confirmed that miR-431 is expressed in PEDS-RMS tumor tissues. Data are expressed as mean ± SEM. **p* < 0.05.

**Figure 3 cancers-11-01380-f003:**
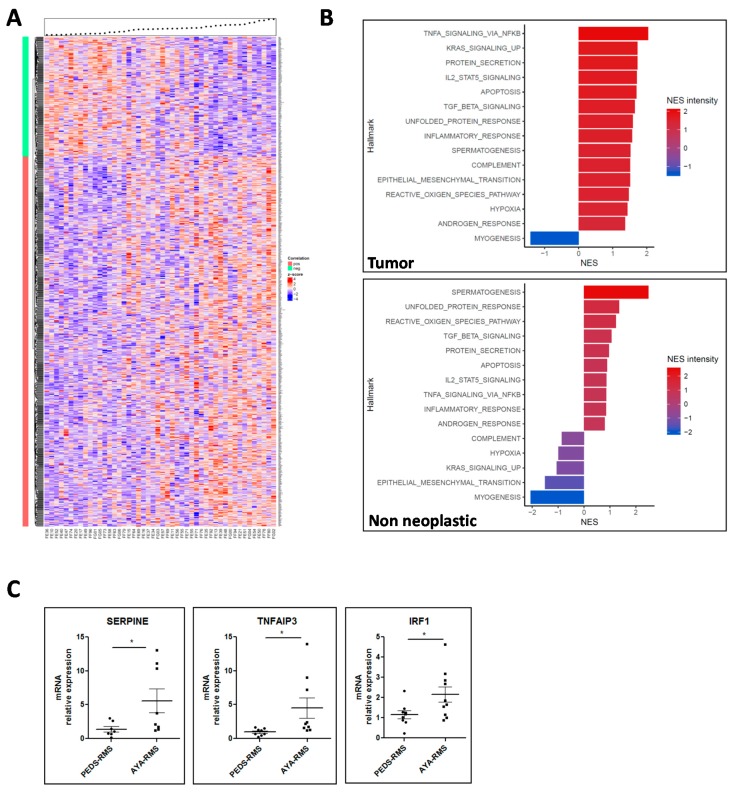
Gene expression modulation in AYA-RMS. (**A)** Heatmap showing the expression of the age-correlated genes according to Spearman’s rank correlation test for tumor samples sorted by age (the black dots above the heatmap). Genes corresponding to the red vertical bar are positively correlated (n = 793) while those associated to the green bar are negatively correlated (n = 194). **(B)** GSEA analysis revealed pathways enrichment according to age in tumor (upper image) and non-neoplastic tissues (lower image). **(C)** Real Time graphs show upregulation of SERPINE, TNFAIP3 and IRF1 in AYA-RMS (n = 10 for each group). Data are expressed as mean ± SEM. **p* < 0.05.

**Figure 4 cancers-11-01380-f004:**
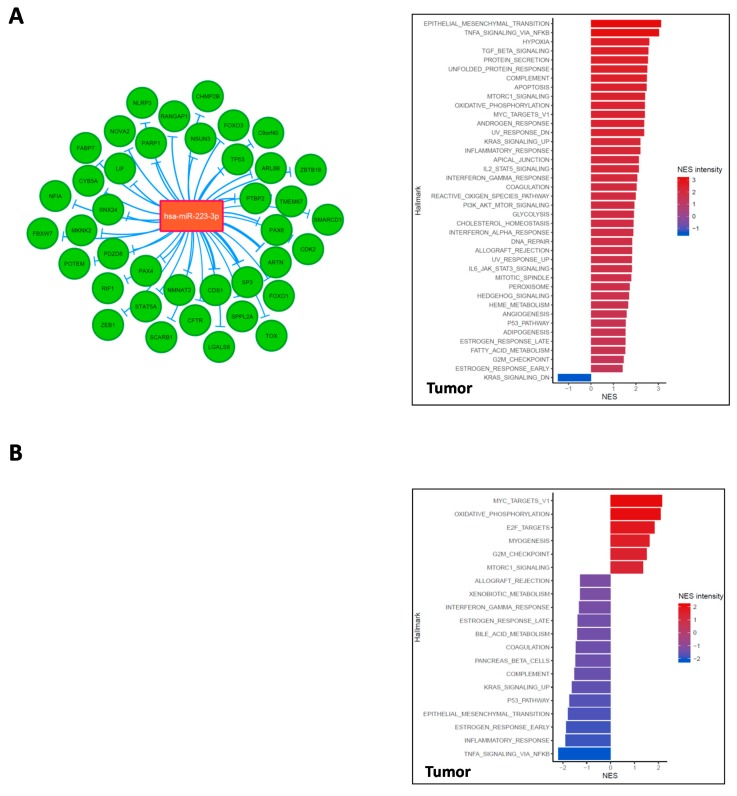
mir-223 and mir-431 regulates several cancer and inflammatory pathways in RMS. (**A**) The figure illustrates an interaction network consisting of our miR-223 (in orange) and their down-modulated target genes (in green) obtained through miRNet in RMS cohort. GSEA graphs show the modulation of several pathways according to miR-223 expression in tumor tissues. (**B**) The figure shows an interaction network consisting of target genes down-modulated (in green) by miR-431(in orange) and their obtained through miRNet. MiR-431 levels modulated several pathways according to GSEA analysis. For both integrated analyses, the size of links between genes and miRNAs differs according to their correlation value: the thicker the link, the more negative is the correlation between the corresponding couple miRNA-gene (that is the more they are anti-correlated).

**Figure 5 cancers-11-01380-f005:**
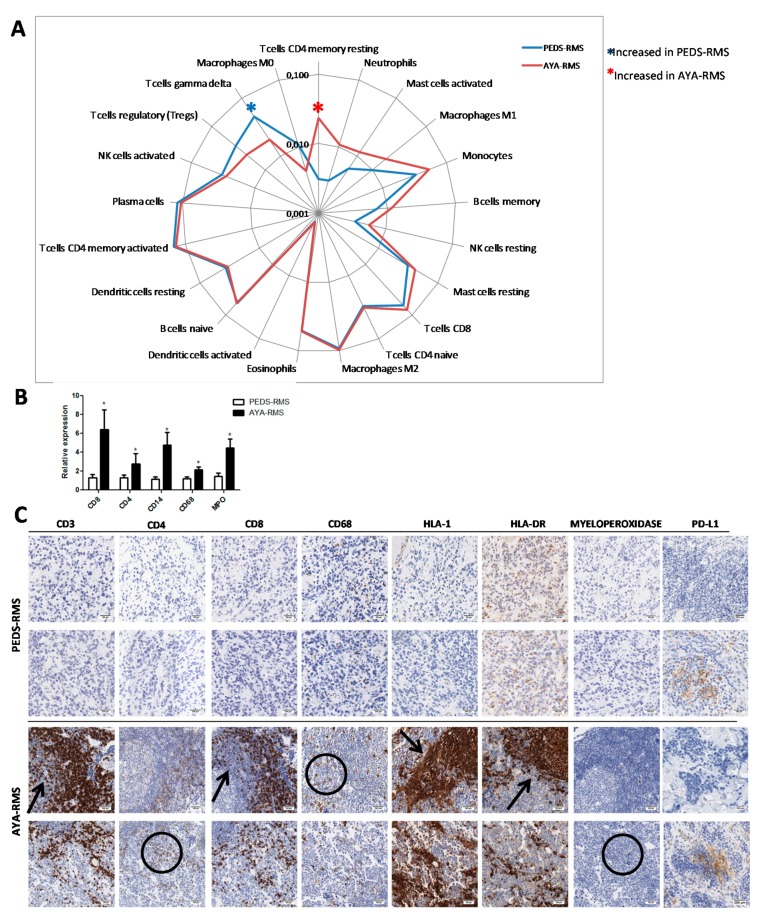
Immune cells invade and infiltrates tumor space in AYA-RMS. (**A**) Radar plot shows enrichment of T cell ϒδ in PEDS-RMS whereas T cell CD4 memory resting in AYA-RMS (n = 49). (**B**) Real Time graphs show an increase of CD4, CD8, CD14, CD68 and MPO in AYA-RMS compared to PEDS-RMS n = 10 for each group). Data are expressed as mean ± SEM. **p* < 0.05 (**C**) Representative images of IHC analysis of different immune cells population (n = 10; original Magnification 200×) in AYA- and PEDS-RMS tumor tissues. CD3, CD8, HLA-1 and HLA-DR are distributed in a pushing-fashion (as indicated by arrows), while positive CD4, CD68 and MPO are intermingled distributed (encircled).

**Figure 6 cancers-11-01380-f006:**
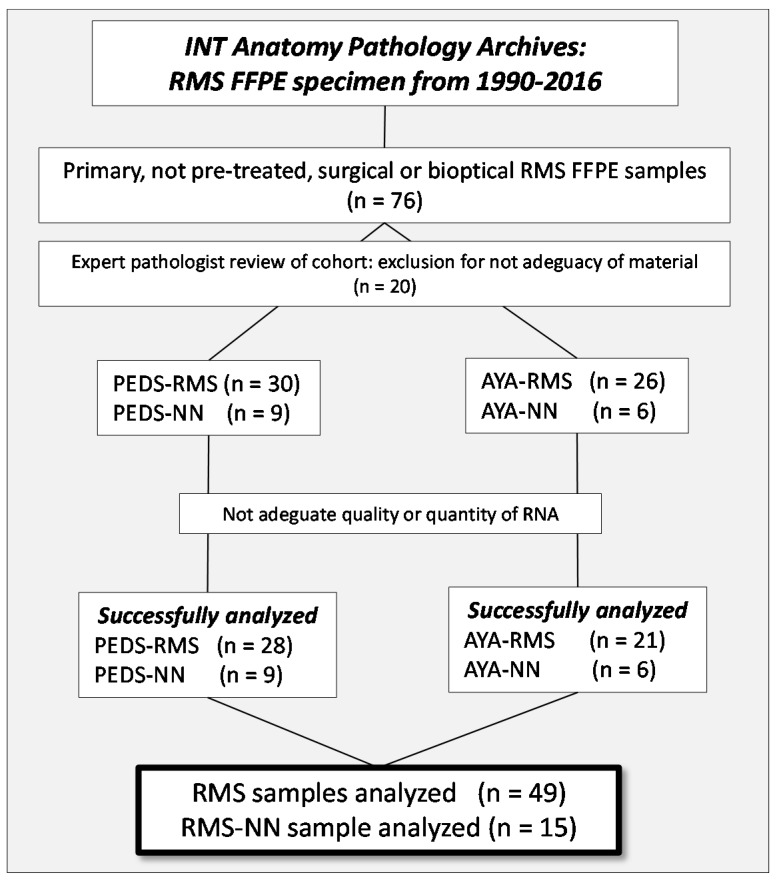
Consort Diagram. A flow diagram of the progress through the phases of recruitment and selection of the studied cohort of PEDS- and AYA RMS.

**Table 1 cancers-11-01380-t001:** Clinical pathological features of the RMS series.

		# of Cases (%)	PEDS 0–14 Years (%)	AYA 15–34 Years (%)
**Age (No. of subjects)**		**49**	**28**	**21**
**Gender**	Male	26 (53)	13 (46)	13 (62)
	Female	23 (47)	15 (54)	8 (38)
**Histology**	Embryonal	29 (59)	20 (71)	9 (43)
	Alveolar	20 (41)	8 (29)	12 (57)
Alveolar Fusion Status	Positive	10	5	5
	Negative	5	3	2
	Not performed	5	0	5
**Primary site**				
Favorable site	Orbits	2 (4)	2 (7)	0 (0)
	Genito-urinary	21 (42)	10 (36)	11 (52)
	Head and Neck	3 (6)	2 (7)	1 (5)
Unfavorable site	Bladder-prostate	2 (4)	0 (0)	2 (9)
	Limbs	7 (14)	6 (21)	1 (5)
	Parameningeal	10 (20)	5 (18)	5 (24)
	Others	4 (8)	3 (11)	0 (0)
	NA	1 (2)	0 (0)	1(5)
**Tumor Stage**	T1	16 (33)	15 (54)	1 (5)
	T2	32 (65)	13 (46)	19 (90)
	NA	1 (2)	0 (0)	1 (5)
**Lymph node involvement**	N0	34 (69)	23 (82)	11 (52)
	N1	14 (28)	5 (18)	9 (43)
	NA	1(2)	0 (0)	1 (5)
**Presence of metastasis**	M0	36 (73)	23 (82)	13 (62)
	M1	12 (25)	5 (18)	7 (33)
	NA	1 (2)	0 (0)	1 (5)
**IRS**	1	9 (18)	5 (18)	4 (19)
	2	4 (8)	1 (4)	3 (14)
	3	23 (47)	17 (60)	6 (29)
	4	12 (25)	5 (18)	7 (33)
	NA	1 (2)	0 (0)	1 (5)
**Overall Survival**	1°CR	25 (51)	16 (57)	9 (43)
	2^°^ CR	2 (4)	2 (7)	0 (0)
	DOD	19 (39)	9 (32)	10 (47)
	Dead of other causes	1 (2)	1 (4)	0 (0)
	NA	2 (4)	0 (0)	2 (9)

NA: Not available; 1° CR: the complete disappearance of disease after first line treatment; 2° CR: the complete disappearance of disease after second line therapy, following tumor relapse; DOD: Dead of other causes; IRS: Intergroup Rhabdomyosarcoma Staging.

## Data Availability

Microarray data, compliant to MIAME (Minimum Information about a Microarray Experiment) guidelines, are available at the Gene Expression Omnibus (GEO) database of NCBI (National Center for Biotechnology Expression) (http://www.ncbi.nlm.nih.gov/geo/) with the accession number GSE135517 (gene-expression) and GSE135518 (miRNA).

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
