# Peer review of "Age-Related Alterations in Immune Contexture Are Associated with Aggressiveness in Rhabdomyosarcoma"

_cancers, 2019, doi:10.3390/cancers11091380_

Round 1

Reviewer 1 Report

Thank you for the opportunity to read and review the manuscript by Dr. Gasparini and colleagues entitled "Age-related alterations in immune contexture drive aggressiveness in rhabdomyosarcoma."  Rhabdomyosarcoma is challenging in the AYA population and the authors are to be commended for tackling such a timely and important issue of determining what the biological differences between AYA and younger patients rhabdomyosarcoma patients might be.  The experiments performed (microarray, miRNA microarray, immunohistochemistry for immune cell infiltrate in tumors) are well done and the literature review is comprehensive.  The conclusions, that miR-223 is overexpressed and miR-431 downregulated in AYA RMS, which is associated with an upregulation of inflammatory pathways and downregulation of muscle development pathways in AYA tumors are well supported by the data presented.  I have several minor points for the authors to consider:

1) the authors should be consistent with the nomenclature for miRs as they appear in the text.  As the document stands sometimes the authors use mir and others miR

2) It is currently well established that the PAX fusion status rather than the histology (alveolar vs embryonal) of the tumor is more prognostic.  There appear to be a large number of Fusion negative alevolar tumors in this cohort.  This should be commented upon in the text.  In addition, in figure 1 a panel of OS for fusion positive versus fusion negative would be helpful. 

3) Recently it has been appreciated that spindle cell/sclerosing RMS, due to mutations in MYOD1, is particularly aggressive and affects predominantly AYA patients.  Do the authors have this histology in their cohort of tumors?  This subtype is also preferentially found in the head and neck.  It might be worthwhile to comment on this in the text and also to separate out the primary site location into more descriptive terms than favorable vs. unfavorable. 

4) It might be worthwhile to also stain tumors for PD1/PDL1 if commenting upon the use of immune checkpoint inhibitors in this population.  Did any patients receive this therapy in the authors' care?  What was the result?

5) There are occasional grammatical errors (conclusion paragraph, for example)

Author Response

Thank you for the opportunity to read and review the manuscript by Dr. Gasparini and colleagues entitled "Age-related alterations in immune contexture drive aggressiveness in rhabdomyosarcoma."  Rhabdomyosarcoma is challenging in the AYA population and the authors are to be commended for tackling such a timely and important issue of determining what the biological differences between AYA and younger patients rhabdomyosarcoma patients might be.  The experiments performed (microarray, miRNA microarray, immunohistochemistry for immune cell infiltrate in tumors) are well done and the literature review is comprehensive.  The conclusions, that miR-223 is overexpressed and miR-431 downregulated in AYA RMS, which is associated with an upregulation of inflammatory pathways and downregulation of muscle development pathways in AYA tumors are well supported by the data presented.  I have several minor points for the authors to consider:

1) the authors should be consistent with the nomenclature for miRs as they appear in the text.  As the document stands sometimes the authors use mir and others miR.

A(nswer): We modified the text accordingly.

2) It is currently well established that the PAX fusion status rather than the histology (alveolar vs embryonal) of the tumor is more prognostic.  There appear to be a large number of Fusion negative alevolar tumors in this cohort.  This should be commented upon in the text.  In addition, in figure 1 a panel of OS for fusion positive versus fusion negative would be helpful. 

A: We agree with the reviewer’s comment regarding the prognostic value of PAX fusion status rather than the histology. Table 1 was appropriately modified in order to have more visible the fusion status numbers of the Alveolar RMS cases. Overall, we have a total of 20 Alveolar RMS, of which 5 fusion positive, 5 fusion negative. Additionally, for 5 ARMS tumors, PAX fusion investigation was not performed as they are older biopsies or extremely small tumors, and we are challenged by insufficient material. We also adapted the manuscript: “Histologically, of 49 RMS, 29 were diagnosed as embryonal while 20 alveolar, of which 10 resulted positive for the PAX fusion transcript, 5 negative and 5 were classified as not performed (insufficient material to investigate their PAX fusion status)”.

Moreover, we modified Figure 1 panel A by adding a Kaplan Maier of the OS for fusion positive vs fusion negatives.

 3) Recently it has been appreciated that spindle cell/sclerosing RMS, due to mutations in MYOD1, is particularly aggressive and affects predominantly AYA patients.  Do the authors have this histology in their cohort of tumors?  This subtype is also preferentially found in the head and neck.  It might be worthwhile to comment on this in the text and also to separate out the primary site location into more descriptive terms than favorable vs. unfavorable. 

A: Our cohort does not include any spindle/sclerosing RMS sample. We improved Table 1 including primary sites. Overall our clinical series includes 3 head and neck RMS (2 in the pediatric and 1 in the adolescents group. 

 4) It might be worthwhile to also stain tumors for PD1/PDL1 if commenting upon the use of immune checkpoint inhibitors in this population.  Did any patients receive this therapy in the authors' care?  What was the result?

We believe that this comment is appropriate and we investigated on the PD-L1 protein expression status in our cohort. Absence of PD-L1 expression was observed in tumor cells whereas immune infiltrating cells were positive in both PEDS-RMS and AYA-RMS samples (Figure 5 C). Since the IHC analysis was performed on a restricted number of  RMS samples, further investigation will be necessary to delineate the role of immunotherapy in RMS clinical management. No pediatric rhabdomyosacoma patient in our Institute received immunotherapy.

5) There are occasional grammatical errors (conclusion paragraph, for example)

A: We modified our manuscript according to revision of three native English speaking colleagues.

Reviewer 2 Report

This manuscript describes a study of the underlying genetic changes on the microRNA level that dictate the differences in aggressiveness between pediatric and AYA rhabdomyosarcoma. Given the lack of data regarding biological differences in AYA RMS versus children, and the uniqueness of these findings, this was a interesting study to add to the literature.   

The paper would benefit from an primary English-language editor. Introduction- Recommend consideration of a global view of outcomes in AYA and a stronger argument as to why biological differences may dictate these outcomes, given that center-based differences, as the paper mentions, may mitigate any survival disadvantage. Recommend including additional introductions of terms such as "oncomiR" and the discussion of myomiRs as a very general introduction to the world of miRNA of cancers and muscles. Methods would benefit from inclusion of Supplementary Figure 4 into the main body if the journal format allows so as to help the reader understand how the samples were obtained in a clear and concise manner. Elaborating on several of the choices made (selection of cases - what were the histologies/fusion status/staging/grouping/metastatic status and if not supplied for manuscript then placed in supplemental data; the selection of certain miR and findings to study and not others) would be extremely helpful for reproducibility of the data. It may be worthwhile to verify these findings with high-throughput sequencing analysis to confirm miRNAs identified. Justification should be provided if the authors think these additional tests are not necessary. Please verify the significance of p-values above 0.05. Please provide clearer figures Recommend additional consideration in the discussion from a clinical vantage as to the translational significance of the immune microenvironment changes identified. How can we theoretically utilize them for therapeutic gain? Do the authors hypothesize if miRNA can be altered for therapeutic gain or followed as prognostic markers? If data permits, would be a good addition to see corelation with patient outcomes and miRNAs identified?

Author Response

This manuscript describes a study of the underlying genetic changes on the microRNA level that dictate the differences in aggressiveness between pediatric and AYA rhabdomyosarcoma. Given the lack of data regarding biological differences in AYA RMS versus children, and the uniqueness of these findings, this was a interesting study to add to the literature.   

The paper would benefit from an primary English-language editor.

A: We modified our manuscript following revision of three native English speaking colleagues.

Introduction- Recommend consideration of a global view of outcomes in AYA and a stronger argument as to why biological differences may dictate these outcomes, given that center-based differences, as the paper mentions, may mitigate any survival disadvantage. A: Thank you for this observation. In our recent study, Bergamaschi et al. reinforced the idea that adherence to the principles of pediatric protocols, improves adult RMS outcomes. Briefly, the measures adopted at our institute (i.e., a multidisciplinary team involving both adult and pediatric oncologists, specific therapeutic recommendations, prospective registration of patients in an institutional database) enabled an increase in the number of adult patients treated according to our strategy for managing pediatric RMS. However, treating adults with pediatric-type strategy is not enough to achieve the results obtained in children. Issues in compliance and a more aggressive biology of adult RMS might have a role in the different outcome according to age. Moreover, we believe that improving the collaboration between pediatric and adult oncologists in promoting specific clinical and biological research is crucial to improve the outcome for this patient population. (Ref 6).

We modified the introduction accordingly: “Treating adults with pediatric-type strategy is not sufficient to achieve similar results obtained in children. Inadequate treatment is certainly an issue but a more aggressive biology of adult RMS might play a pivotal role in the different outcome according to age.”

Recommend including additional introductions of terms such as "oncomiR" and the discussion of myomiRs as a very general introduction to the world of miRNA of cancers and muscles. A: As suggested, we implemented the introduction section with a better explanation of “oncomiR”.

“miRNAs are deregulated in different disease such as cancer, hepatitis and metabolic disease [10]. Based on their function, miRNAs may be considered as “oncomir” when their over-expression in tumor inhibits a tumor suppressor gene, for instance mir-17-92 cluster in lung cancer [1]. On the other hand, a “tumor suppressor miRNA” such as let-7a is down-regulated in cancer cells and its loss causes aberrant expression of an oncogene [2]. Evidence reports that both myomiRs, which are muscle enriched miRNAs, and non-myomiRs, which are expressed in muscles, but also in other tissues, play a critical role in RMS development, progression, and could potentially be therapeutic targets for RMS [11,12].”

We also included a short discussion on myomiRs.

Furthermore, miR-206 was described as a skeletal muscle miRNA that controlled proliferation of muscle cells through Pax7 downregulation [3]. Regarding non-myomiRs miR-17-92 cluster is over-expressed in RMS and could be responsible for RMS aggressiveness [4]. Another miR linked to metastatic dissemination of RMS is miR-185 that modulates Six1 [5]. While gene expression profiling (GEP) in RMS  provides survival prediction signatures and define molecular classes [13,14], no studies focused on the description of age-related biological targets and pathways have been so far performed.”

 Methods would benefit from inclusion of Supplementary Figure 4 into the main body if the journal format allows so as to help the reader understand how the samples were obtained in a clear and concise manner.

A: We agree with this recommendation and we added supplementary figure 4 as Figure 6 within the manuscript.

 Elaborating on several of the choices made (selection of cases - what were the histologies/fusion status/staging/grouping/metastatic status and if not supplied for manuscript then placed in supplemental data; the selection of certain miR and findings to study and not others) would be extremely helpful for reproducibility of the data.

A: As for the selection of RMS cases, our cohort comprise consecutive RMS treated at our Institution. Main restrictions of inclusion were: sample availability and adequacy, tumors were to be primary and not pre-treated. These restrictions lead us to exclude plenty RMS: a) lots of pre-treated RMS; extremely small biopsies with insufficient material for the analyses; c) RMS samples prior of 1999, inadequate as fixed in buoin that makes RNA extraction challenging.

For this we appropriately adapted the manuscript: “The study is a monocentric retrospective analysis of a cohort of consecutive cases diagnosed with RMS and treated in our Institute, were recovered from our Institutional series between the years 1999 and 2016 at the Fondazione IRCCS of the Istituto Nazionale dei Tumori (INT). Seventy-six Formalin Fixed Paraffin Embedded (FFPE) tissues of primary, non-pre-treated, surgical or bioptic specimen of PEDS- and AYA-RMS were selected and retrieved from the archives of the Department of Diagnostic Pathology and from the Laboratory of Medicine of our Institute. Moreover, MiRNA validation was performed by selecting miRNAs with a described role in muscle development or involved in the modulation of muscular cell phenotype.

It may be worthwhile to verify these findings with high-throughput sequencing analysis to confirm miRNAs identified. Justification should be provided if the authors think these additional tests are not necessary.

A: Thank you for the suggestion. For the purpose of this study, we validated our miRNAs data with Real Time PCR so to be able to confirm miRNAs microarray findings. We are not able to perform another high-throughput sequencing such as RNAseq due to the high costs of this technology and also to the small amount of RNA left in the study cohort.

 Please verify the significance of p-values above 0.05.

A: We added the p-values significance as requested.

 Please provide clearer figures

 A: As requested we provide figures with high resolution.

 Recommend additional consideration in the discussion from a clinical vantage as to the translational significance of the immune microenvironment changes identified. How can we theoretically utilize them for therapeutic gain? Do the authors hypothesize if miRNA can be altered for therapeutic gain or followed as prognostic markers? If data permits, would be a good addition to see correlation with patient outcomes and miRNAs identified? 

A: We thank the reviewer for these interesting considerations. Since we observed an increase of immune cell infiltration in AYA-RMS compared to PEDS we speculate that therapeutic strategies targeting immune cell recruitment or modulating the activation of these cells combined with standard drugs could be useful for the management of RMS. Further studies will be done to assess the functional role of miRNAs (in particular mir-223 and mir-431) in RMS development. Since the small number of cases in our cohort we didn’t observed any statistical significant correlation between miRNAs identified and OS of patients.

 Reference List

Osada H, Takahashi T (2011) let-7 and miR-17-92: Small-sized major players in lung cancer development. Cancer Sci 102: 9-17. Lee YS, Dutta A (2007) The tumor suppressor microRNA let-7 represses the HMGA2 oncogene. Genes Dev 21: 1025-1030. Taulli R, Bersani F, Foglizzo V, Linari A, Vigna E et al. (2009) The muscle-specific microRNA miR-206 blocks human rhabdomyosarcoma growth in xenotransplanted mice by promoting myogenic differentiation. J Clin Invest 119: 2366-2378. Reichek JL, Duan F, Smith LM, Gustafson DM, O'Connor RS et al. (2011) Genomic and clinical analysis of amplification of the 13q31 chromosomal region in alveolar rhabdomyosarcoma: a report from the Children's Oncology Group. Clin Cancer Res 17: 1463-1473. Imam JS, Buddavarapu K, Lee-Chang JS, Ganapathy S, Camosy C et al. (2010) MicroRNA-185 suppresses tumor growth and progression by targeting the Six1 oncogene in human cancers. Oncogene 29: 4971-4979.

Reviewer 3 Report

The study is a clinical correlative analysis of miRNA profiles and gene expression in a cohort of AYA-RMS and PED-RMS patients from a single institution. The background information states that AYA-RMS patients have inferior outcomes when compared to PED-RMS patients because of a higher prevalence of clinically unfavorable features and heterogeneity in treatment regimens administered to older patients. AYA-RMS patients were less likely to be treated on cooperative protocols. In addition, the authors hypothesize that tumor biology may be different between AYA-RMS and PED-RMS and aimed to investigate this in the current study. The results indicate that miR-223 overexpression and mir-431 down regulation were found in AYA-RMS when compared to PED-RMS. In addition, AYA-RMS tumors were found to have increased CD4 T-cell, CD8 T-cell, and neutrophil infiltration when compared to PED-RMS. In general, the study is well-designed and the conclusions are supported by the results.  I suggest the following revisions prior to publication of this manuscript in Cancers.

MAJOR:

Because this is primarily a descriptive study in which functional experiments were not performed, I do not think the data support the title. I recommend changing the title to “Age-related alterations in immune contexture are associated with aggressiveness in rhabdomyosarcoma”. In Table 1 the age range for AYA-RMS is listed as 15-30 years and in Figure 1 the age range for AYA-RMS is listed as 15-34 years. Why is there a discrepancy? In Table 1 primary and secondary complete responses are not defined. In Table 1, it states that there are 12 patients with alveolar histology, but only 4 patients with a fusion transcript. Do the authors think that AYA-RMS patients are more likely to have fusion-negative tumors with alveolar histology? Might be worth considering. The Figure legend for Figure 1 doesn’t state that the result is no difference in histology between PED-RMS or AYA-RMS. I would state this in the Figure legend to help the reader understand what you are trying to show. Figure 1 needs scale bars for the histology images. Throughout the manuscript, state the actual p-values rather than “p<0.05” The results section (Figure 2 B) states that miR-223 is associated with FOXO signaling. Is there a difference in miR-223 expression when fusion positive versus fusion negative RMS are compared in this cohort? Figure 1 – This may be an artifact of the *.pdf file used for review, but all the text in this figure (A,B,C) is too small and too blurry to interpret. The heatmap legend is too small to interpret. I would also indicate where mir-223 and mir-431 are on the heatmap. I suggest using a higher resolution graphic so that the reader can zoom in on the figure and be able to see each miRNA of interest to them. The same thing is true for Figure 3 A (needs a higher resolution graphic and increased legend size). The supplementary figures have excellent resolution, so I assume that high resolution graphics are available for inclusion in the main manuscript. Figure 4 – It is impossible to see blue text on a green background and therefore, the reader cannot interpret the graphics depicting the interaction networks in their current format. Line 159 states that EMT pathways correlated with miR-223 levels suggesting that regulation of tumor suppressor Fbxw7 could be responsible for this pro-tumorigenic effect. This seems rather speculative. I do not see where a specific role for FBXW7 is supported by Figure 4. Lines 179-180 – I don’t think gene expression analysis can quantify specific tumor-infiltrating immune cells, but rather can quantify gene expression associated with these cells. Figure 5 C is an excellent figure, but it needs scale bars. In line 242 it states that the integrated data show that miRNA is crucial for myogenesis mechanisms. I would say that the current study shows an association, but not a critical relationship without specific functional experiments. In line 262-262 the authors definitively state that the outcomes of AYA-RMS are better at their institution because of adherence to a consistent clinical management strategy. While this may be true, I feel that the way it is phrased is too definitive. I would tone down the way this is stated to include it as a possibility, but not a definite reason. In the discussion section, the authors speculate that immune-checkpoint inhibitor blockade could be an effective option for AYA-RMS. Please quantify the expression of PD-1, PD-L1, and CTLA-4 in AYA-RMS versus PED-RMS in this study. This could be an exciting future direction.

MINOR

The grammar in the methods section is impeccable, but the grammar throughout the remainder of the manuscript could use some work. I suggest the manuscript be submitted to an English language scientific editor prior to resubmission. Please define AYA-RMS and PED-RMS clearly (including age ranges) in the introduction section of the manuscript. Please standardize the way that AYA-RMS and PED-RMS are displayed throughout the manuscript. There are some areas where they are unhyphenated and others where a hyphen is utilized, please be consistent. The abstract states that gene expression profiling (GEP) showed an increase in CD4 memory resting cells and a decrease in T-cells in AYA-RMS. I do not think GEP is capable of definitively showing this. However, this was shown in the manuscript by immunohistochemistry. I would say that GEP showed an increase in genes associated with these cells rather than the cells themselves. In the introduction section, the authors state that muscle specific miRNAs are used as a therapeutic tool in RMS. I believe the work showing this is all preclinical and I would therefore not state that they are being utilized as a therapeutic tool. Please verify. In the results section line 91 it states that “As expected, PED-RMS revealed a better outcome even if not statistically significant.” I would avoid the use of language like this. Maybe just say that a nonsignificant separation of Kaplan Meier survival curves was observed between PED-RMS and AYA-RMS patients. Table 1 – “Lymphnode involvement” needs a space “Lymph node involvement” In Figure 2 and 3 legends, the number of negatively correlated genes or miRNAs are listed, but the number for the positively correlated genes or miRNAs is not listed. In the methods section line 328 it says “with the exclusion of a RMS tumor”. What does this mean?

Author Response

The study is a clinical correlative analysis of miRNA profiles and gene expression in a cohort of AYA-RMS and PED-RMS patients from a single institution. The background information states that AYA-RMS patients have inferior outcomes when compared to PED-RMS patients because of a higher prevalence of clinically unfavorable features and heterogeneity in treatment regimens administered to older patients. AYA-RMS patients were less likely to be treated on cooperative protocols. In addition, the authors hypothesize that tumor biology may be different between AYA-RMS and PED-RMS and aimed to investigate this in the current study. The results indicate that miR-223 overexpression and mir-431 down regulation were found in AYA-RMS when compared to PED-RMS. In addition, AYA-RMS tumors were found to have increased CD4 T-cell, CD8 T-cell, and neutrophil infiltration when compared to PED-RMS. In general, the study is well-designed and the conclusions are supported by the results.  I suggest the following revisions prior to publication of this manuscript in Cancers.

Major:

Because this is primarily a descriptive study in which functional experiments were not performed, I do not think the data support the title. I recommend changing the title to “Age-related alterations in immune contexture are associated with aggressiveness in rhabdomyosarcoma”.

A: We modified the title as suggested.

 In Table 1 the age range for AYA-RMS is listed as 15-30 years and in Figure 1 the age range for AYA-RMS is listed as 15-34 years. Why is there a discrepancy?

A: We made a mistake: in Table 1 we were general and listed 15-+30 but to be precise it is 15-34. For this reason we corrected the discrepancy in Table 1 with 15-34 to be in concordance with Figure 1.

 In Table 1 primary and secondary complete responses are not defined.

A: We added the definitions below Table 1

     “1° CR –the complete disappearance of disease after first line treatment; 2° CR the complete disappearance of disease after second line therapy, following tumor relapse”

In Table 1, it states that there are 12 patients with alveolar histology, but only 4 patients with a fusion transcript. Do the authors think that AYA-RMS patients are more likely to have fusion-negative tumors with alveolar histology? Might be worth considering.

A: We agree with the reviewer’s comment and we realize that the way we stated the number of cases in Table 1 was a bit confusing. Therefore, Table 1 was appropriately modified in order to have more visible the fusion status numbers of the Alveolar RMS cases. Overall, we have a total of 20 Alveolar RMS, of which 5 fusion positive, 5 fusion negative. In 5 ARMS, PAX fusion investigation was not performed as they are older biopsies or extremely small tumors, and we are challenged by insufficient material. We also adapted the manuscript: “Histologically, of 49 RMS, 29 were diagnosed as embryonal while 20 alveolar, of which 10 resulted positive for the PAX fusion transcript, 5 negative and 5 were classified as not performed (insufficient material to investigate their PAX fusion status)”.

Additionally, we modified Figure 1 panel A by adding a Kaplan Maier of the OS for fusion positive vs fusion negatives.

 The Figure legend for Figure 1 doesn’t state that the result is no difference in histology between PED-RMS or AYA-RMS. I would state this in the Figure legend to help the reader understand what you are trying to show.  

A: We modified the Figure legend as requested.

Figure 1 needs scale bars for the histology images.

A: We added the scale bars.

Throughout the manuscript, state the actual p-values rather than “p<0.05”

A: We modified the p-value as requested and added Supplementary tables for miRNAs and genes.

The results section (Figure 2 B) states that miR-223 is associated with FOXO signaling. Is there a difference in miR-223 expression when fusion positive versus fusion negative RMS are compared in this cohort?

A: As requested, we analyzed miR-223 considering Fusion stated but we didn’t observe any difference Average (2^-dCT Fusion positive vs negative 1.77 vs 2.80).

Figure 1 – This may be an artifact of the *.pdf file used for review, but all the text in this figure (A,B,C) is too small and too blurry to interpret. The heatmap legend is too small to interpret. I would also indicate where mir-223 and mir-431 are on the heatmap. I suggest using a higher resolution graphic so that the reader can zoom in on the figure and be able to see each miRNA of interest to them. The same thing is true for Figure 3 A (needs a higher resolution graphic and increased legend size). The supplementary figures have excellent resolution, so I assume that high resolution graphics are available for inclusion in the main manuscript. Figure 4 – It is impossible to see blue text on a green background and therefore, the reader cannot interpret the graphics depicting the interaction networks in their current format.

A: As suggested we provided and modified figures with high resolution. 

Line 159 states that EMT pathways correlated with miR-223 levels suggesting that regulation of tumor suppressor Fbxw7 could be responsible for this pro-tumorigenic effect. This seems rather speculative. I do not see where a specific role for FBXW7 is supported by Figure 4.

 A: I agree with the reviewer that since we did not investigate on the role of FBXW7 rather we were simply speculating. Therefore we changed the sentenced accordingly: “Three genes resulted the most anti-correlated: CDS1, SP3 and ARTN. In RMS tumor tissues, integration analysis showed an up-regulation of EMT pathways correlated with miR-223 levels speculating that regulation of tumor suppressor Fbxw7 could be responsible of this pro-tumorigenic effect.”

Lines 179-180 – I don’t think gene expression analysis can quantify specific tumor-infiltrating immune cells, but rather can quantify gene expression associated with these cells.

A: It is an appropriate comment and we reworded the sentence: “Since MiRNAs and GEP investigations highlight the relevance of inflammation and immune contexture in AYA- and PEDS-RMS, we utilized CIBERSORT, an analytical tool developed by Newman et al., to provide an estimation of the abundances of the immune composition of RMS tumor sample using gene expression data.”

Figure 5 C is an excellent figure, but it needs scale bars.

A: We added the scale bars.

In line 242 it states that the integrated data show that miRNA is crucial for myogenesis mechanisms. I would say that the current study shows an association, but not a critical relationship without specific functional experiments.

A: The sentence was properly modified: Our integrated data shows an association of this miRNA with myogenesis mechanisms, supporting literature that describes the modulation of myostatin and muscle growth and regeneration by miR-431 [25].”

In line 262-262 the authors definitively state that the outcomes of AYA-RMS are better at their institution because of adherence to a consistent clinical management strategy. While this may be true, I feel that the way it is phrased is too definitive. I would tone down the way this is stated to include it as a possibility, but not a definite reason.

A: We agree with the reviewer and modified the sentence accordingly: Moreover, although clinical information was available for our cohort, the differences in overall survival were not significant between the two groups.  One possible explanation could be related to the clinical management of AYA patients in our Institution, intended as different chemotherapy approaches and tumor-associated factors specific for this peculiar age group.”

In the discussion section, the authors speculate that immune-checkpoint inhibitor blockade could be an effective option for AYA-RMS. Please quantify the expression of PD-1, PD-L1, and CTLA-4 in AYA-RMS versus PED-RMS in this study. This could be an exciting future direction.

A: In agreement with the recommendation by the reviewer, we investigated the PD-L1 status on a restricted RMS cohort by IHC. Our data showed absence of PD-L1 expression in the tumor whereas immune infiltrating cells were similarly positive for both AYA-RMS and PEDS-RMS. We added in the conclusion section: ”Notably, PD-L1 expression in immune infiltrating cells of both PEDS and AYA-RMS suggest the possibility to target this immune checkpoint in selected patients. Further investigations are needed to better clarify the use of immune checkpoints inhibitors, combined with novel immunotherapeutic strategies to block the recruitment of immune for the treatment of this aggressive disease.”

Minor

The grammar in the methods section is impeccable, but the grammar throughout the remainder of the manuscript could use some work. I suggest the manuscript be submitted to an English language scientific editor prior to resubmission.

Please define AYA-RMS and PED-RMS clearly (including age ranges) in the introduction section of the manuscript. A: We defined the age groups including the age-ranges modifying a sentence: “The epidemiological EUROCARE-5 working group (study period: 2000–2007) reported a 6% 5-year overall survival among 0–14 years old patients (PEDS-RMS), as opposed to 38% for patients 15–39 years of age (AYA-RMS)”.

 Please standardize the way that AYA-RMS and PED-RMS are displayed throughout the manuscript. There are some areas where they are unhyphenated and others where a hyphen is utilized, please be consistent.

A: We standardized AYA-RMS and PEDS-RMS throughout the manuscript. The abstract states that gene expression profiling (GEP) showed an increase in CD4 memory resting cells and a decrease in T-cells in AYA-RMS. I do not think GEP is capable of definitively showing this. However, this was shown in the manuscript by immunohistochemistry. I would say that GEP showed an increase in genes associated with these cells rather than the cells themselves. A: We agree with this comment and modified the sentence: “GEP showed an increase in genes associated with CD4 memory resting cells and a decrease in genes associated with γδ T-cells in AYA-RMS”. In the introduction section, the authors state that muscle specific miRNAs are used as a therapeutic tool in RMS. I believe the work showing this is all preclinical and I would therefore not state that they are being utilized as a therapeutic tool. Please verify. A: We verified and we adequately reworded the sentence: “Evidence reports that both myomiRs, which are muscle enriched miRNAs, and non-myomiRs, which are expressed in muscles, but also in other tissues, play a critical role in RMS development, progression, and could potentially be therapeutic targets for RMS [11,12]. In particular, miR-22, a miRNA induced in physiological normal muscle differentiation, is down-regulated in RMS and its replacement blocked tumor growth and dissemination in pre-clinical in vivo models [12].”

In the results section line 91 it states that “As expected, PED-RMS revealed a better outcome even if not statistically significant.” I would avoid the use of language like this. Maybe just say that a nonsignificant separation of Kaplan Meier survival curves was observed between PED-RMS and AYA-RMS patients.

A: Thank you for the suggestion and we adapted the sentence: Non-significant separation of Kaplan Meier survival curves was observed between PED-RMS and AYA-RMS patients.”

Table 1 – “Lymphnode involvement” needs a space “Lymph node involvement”.

A: We modified the table as suggested

 In Figure 2 and 3 legends, the number of negatively correlated genes or miRNAs are listed, but the number for the positively correlated genes or miRNAs is not listed.

A: As suggested, we added figure legends adding the number of correlated miRNAs and genes.

In the methods section line 328 it says “with the exclusion of a RMS tumor”. What does this mean? 

A: Thank you for the comment. The phrasing of this sentence is surely not clear. What was intended is that while miRNA profiling was performed on 64 samples of total RNA, GEP analysis was performed on 63 samples. Total RNA extracted was not sufficient for one RMS sample to perform both miRNA and GEP analysis. As suggested we modified the sentence: “ Total RNA from 63 RMS samples, the same material used in miRNA profiling, was reverse-transcribed, amplified, fragmented, biotin-labeled and then hybridized to the Affymetrix GeneChip Human Clariom S Array (ThermoFisher Scientific). One RMS specimen was excluded from the analysis for insufficient quantity of total RNA.